# Design of a Semiactive TMD for Lightweight Pedestrian Structures Considering Human–Structure–Actuator Interaction

Christian A. Barrera-Vargas [1,*], Javier Naranjo-Pérez [1], Iván M. Díaz [1] and Jaime H. García-Palacios [2]

1 Department of Continuum Mechanics and Theory of Structures, ETS Ingenieros Caminos, Canales y Puertos, Universidad Politécnica de Madrid, 28040 Madrid, Spain; javier.naranjo@externos.upm.es (J.N.-P.); ivan.munoz@upm.es (I.M.D.)
2 Department of Hydraulics, Energy and Environmental Engineering, ETS Ingenieros Caminos, Canales y Puertos, Universidad Politécnica de Madrid, 28040 Madrid, Spain; jaime.garcia.palacios@upm.es
* Correspondence: christian.barrera.vargas@alumnos.upm.es

**Abstract:** Lightweight pedestrian structures constructed with high strength-to-weight ratio materials, such as fiber-reinforced polymers (FRP), may experience large accelerations due to their lightness, thus overcoming the serviceability limit state. Additionally, uncertainties associated with human–structure interaction phenomena become relevant. Under these circumstances, variations in pedestrian actions could modify the modal properties of the coupled human–structure system and classical approaches based on passive Tuned Mass Dampers (TMD) do not offer an effective solution. An alternative solution is to use a Semiactive TMD (STMD), which includes a semiactive damper that, when properly designed, may be effective for a relatively broad frequency band, offering a robust solution when significant uncertainties are present. Thus, this paper presents a design methodology for the design of STMDs applied to lightweight pedestrian structures including human–structure and actuator–structure interaction. A multiobjective optimization procedure has been proposed to simultaneously minimize structure acceleration, inertial mass, and maximum damper force. The methodology has been applied to a lightweight FRP footbridge. Realistic simulations, including system uncertainties, interaction phenomena, nonlinear damper model, noise-contaminated signals, and the practical elements (in-line digital filters) needed for the successful implementation of the control law, validate the methodology. As a conclusion, the STMD is more effective than its passive counterpart in both, canceling the response or achieving similar performance with significant lower inertial mass.

**Keywords:** human-induced vibrations; semiactive actuator; human–structure interaction; lightweight structures; magnetorheological damper

## 1. Introduction

The use of lightweight materials, such as fiber-reinforced polymers (FRP), in the design and construction of pedestrian footbridges has recently attracted increased interest. This is motivated by several of their benefits, such as durability, high strength-to-weight ratio, and low maintenance cost. Although modal damping ratios of FRP footbridges have been identified to be higher than those of footbridges constructed by conventional materials, the vibration level could exceed the comfort limits defined by design codes and guidelines due to their low effective generalized modal masses [1], even under high and less-energetic harmonics of pedestrian actions.

Lightweight pedestrian structures are more prone to suffer from resonance effects induced by pedestrian actions and the human–structure interaction (HSI) phenomenon becomes a crucial aspect to be considered. Gallegos et al. [2] studied the dynamic behavior of an ultra-lightweight FRP footbridge with excessive vertical vibrations, using a mass-spring–damper–actuator (MSDA) system to model the pedestrian load crossing the footbridge. Díaz et al. [3] presented the interaction phenomena involved when controlling

human-induced vibrations in lightweight structures and their consideration in the frequency domain. The closed-loop transfer functions for structures controlled with strategies based on passive and active Tuned Mass Dampers (TMD) including HSI were presented.

In order to reduce the excessive vibrations, the incorporation of a TMD is an excellent solution to cancel vibrations when its parameters are tuned to a particular vibration mode of the structure [4,5], whose modal properties do not change significantly [6]. Otherwise, detuning problems may arise and more robust TMD-based strategies should be employed.

The uncertainties associated with the estimated modal properties, the excitation of more than one vibration mode, and even the non-negligible nonresonant harmonic response, usually lead to poor performance of passive TMDs when dealing with lightweight structures. Hence, the implementation of more robust strategies to improve the performance of the TMD should be considered; for example, (i) Multiple TMDs (MTMDs) configured in series or parallel, or (ii) a semiactive Tuned Mass Damper (STMD), which includes a semiactive device in the TMD [7]. Van Nimmen et al. [8] studied the experimental response of a slender steel footbridge with and without MTMDs, reporting variations in the results when these were compared with the design procedure detailed in the guidelines, obtaining a lower acceleration response with the implementation of the MTMD. Caetano et al. [9] conducted an experimental study on the implementation of MTMDs to control the vertical vibration and the lock-in effects (lateral vibration) of an in-service footbridge. Regarding Semiactive TMDs (STMDs), most of the applications are based on using a Magnetorheological (MR) damper as a semiactive device [10]. Hence, the use of this type of device requires a control law to govern its behavior. Regarding phase control approaches, Koo et al. [11] proposed a control law for the application within STMDs, which depends on the phase between the displacement of the main structure and the relative velocity between the structure and the TMD inertial mass. Later, Moutinho et al. [12] modified Koo's control law, substituting the displacement by the acceleration and neglecting the structure's velocity with respect to the TMD mass velocity. Thus, this control law is clearly geared to the practice implementation. Other approaches, based on optimal robust controllers, may be found in the literature. However, these approaches are based on the knowledge of the whole system state and are usually difficult to implement in practice, especially when it comes to civil structures with high uncertainty. As an example, Zhang et al. [13] proposed a robust algorithm applied to a semiactive base isolation system. The algorithm combines a linear quadratic regulator and a nonlinear robust compensator to control the oil pressure in the cavity of a friction pendulum system. On the other hand, Gu et al. [14] used an optimal neuro-fuzzy logic control to modify the stiffness of a magnetorheological elastomer, which was also used in a base isolation system.

Based on the results of several numerical studies, the semiactive control device has been demonstrated to outperform the passive device, either in terms of the dynamic response of the structure [15–17] or the size of the device [18]. Several experimental implementations of semiactive control devices have been successfully achieved for human-induced vibrations [19–21]. Nevertheless, there are often important differences between simulation and experimental results. Thus, before sizing and implementing the STMD, realistic simulations should be undertaken, especially when dealing with lightweight structures that may experience HSI. The following issues should be considered: (i) the nonlinear MR damper model; (ii) sensor and electrical noise should be included in order to consider noise-contaminated sensor measurements that may affect the control law performance; (iii) the measured-signals should be filtered according to the frequency band of interest, avoiding phase delays and removing undesirable frequency contents; (iv) a trigger rule to activate/deactivate the damper device becomes essential to avoid, firstly, unstable behavior due to the nonlinear on–off phase control law [22] and, secondly, the unnecessary continuous operation of the damper.

This paper presents a methodology for the optimum design of a STMD using a semiactive damper controlled by a phase-control law. The proposed design methodology considers, for the first time, the human–structure–actuator interaction, which is an im-

portant issue when dealing with lightweight pedestrian structures. Pedestrians react to perceptible ground vibrations by modifying their gait and the HSI phenomenon arises, modifying substantially the coupled system (coupled human–structure model) to be controlled. Additionally, the motion of the lightweight structure may affect significantly the motion of the STMD inertial mass, and this effect can no longer be neglected when predicting the STMD performance and should be considered in the design. The methodology has been applied to the lightweight laboratory FRP footbridge recently constructed by the authors of the paper. The footbridge has been designed in order to fulfill all the structural limit states except the vibration serviceability one, which is expected to be fulfilled by implementing a STMD. Thus, the STMD is designed to meet the vibration serviceability considering system uncertainty and human–structure–actuator interaction. The proposed design methodology makes use of a multiobjective constrained optimization problem that minimizes simultaneously three objective functions. The constraint of the problem allows fulfilling the vibration serviceability limit state and the objective functions are based on the acceleration of the structure, the mass of the control device (directly related to the device cost), and the size of the semiactive device (through its saturation force). Finally, the optimum STMD with a MR damper is analyzed considering a real pedestrian input, noise in the signals of the control law, digital filters for the signals of the control law, and a trigger rule to activate/deactivate the MR damper.

This paper is structured as follows. First, the coupled human–structure system controlled with passive and semiactive TMD-based strategies is explained. Next, Section 3 focuses on the optimum design of the control device, considering uncertainty conditions in both the structure and the pedestrian load. In Section 4, the proposed methodology is applied to the optimum design of a TMD and a viscous STMD to control the dynamic response of a full-scale laboratory FRP footbridge. The STMD including a MR damper model is designed following the proposed methodology under real conditions in Section 5. Finally, the main concluding remarks and suggestions for future work are drawn in Section 6.

## 2. System Modeling

This section describes the theoretical approach considered here for the control of human-induced vibrations. First, the HSI model is detailed. Afterwards, both passive and semiactive control strategies are presented.

### 2.1. HSI Model

The HSI model considers the human interactive force originated from the structure movement. Based on Dougill et al. [23], the dynamic properties of the pedestrian could be analyzed as a MSDA system, while the structure is represented as a Single Degree of Freedom (SDOF) system, for the sake of clarity, defined through the following equation:

$$m_s \ddot{x}_s + c_s \dot{x}_s + k_s x_s = F_h,\tag{1}$$

where the parameters $m_s, c_s, k_s$ are the generalized modal mass, damping coefficient, and modal stiffness of the structure, respectively; $x_s$ is the structure displacement (dots mean time derivatives); and $F_h$ is the resultant force acting on the structure. This system can also be represented in the Laplace domain by the following transfer function (TF) between the structure acceleration and the force:

$$G_S(s) = \frac{s^2 X_s(s)}{F_h(s)} = \frac{s^2}{s^2 m_s + s c_s + k_s},\tag{2}$$

where $s = j\omega$ is the Laplace variable with $\omega$ being the angular frequency (rad/s).

The equations of motion that govern the HSI model are obtained from the force balance of Figure 1a [3] and are given by

$$m_h \ddot{x}_h + c_h(\dot{x}_h - \dot{x}_s) + k_h(x_h - x_s) = F_a, \tag{3}$$

$$m_s \ddot{x}_s + c_s \dot{x}_s + k_s x_s - c_h(\dot{x}_h - \dot{x}_s) - k_h(x_h - x_s) = -F_a, \tag{4}$$

where the subindexes $h$ and $s$ refer to the human and structure, respectively. The human model is defined by the effective body mass ($m_h$, which is a fraction of the total human body mass), the stiffness ($k_h$), the damping coefficient ($c_h$), and the actuator force $F_a$—also known as driving force—that represents a pair of action–reaction forces generated by the human legs that act simultaneously on both the structure and the human. Additionally, the transmitted interaction force, denoted as $F_{hsi}$, is derived as follows:

$$F_{hsi} = c_h(\dot{x}_h - \dot{x}_s) + k_h(x_h - x_s). \tag{5}$$

Taking Laplace transforms in Equations (3)–(5) and considering Equation (2), the following TFs are derived:

$$G_{HSI}(s) = \frac{F_{hsi}(s)}{s^2 X_s(s)} = \frac{-m_h(sc_h + k_h)}{s^2 m_h + sc_h + k_h}, \tag{6}$$

$$G_A(s) = \frac{F_{ha}(s)}{F_a(s)} = \frac{-s^2 m_h}{s^2 m_h + sc_h + k_h}, \tag{7}$$

with $G_{HSI}(s)$ being the TF between the human interacting force and the acceleration response of the structure and $G_A(s)$ being the TF between the human force—without considering the movement of the structure—and the driving force. Using Equations (2), (6) and (7), the HSI model for nonmoving actions (such as jumping or bouncing) can be represented as the block diagram of Figure 1b. To account for a pedestrian passing along the footbridge, the block diagram of Figure 1b can be modified according to the position of the pedestrian, as depicted in Figure 1c. The force acting on the structure must be scaled by the modal shape $\varphi(x)$. Thus, the resultant force acting on the structure will be $\varphi(x)F_h$, in which $x = vt$ and the velocity of the MSDA system ($v$) depends on the gait frequency. Finally, it is worth mentioning that more than one vibration mode can be considered at the control point by modifying $G_S(s)$.

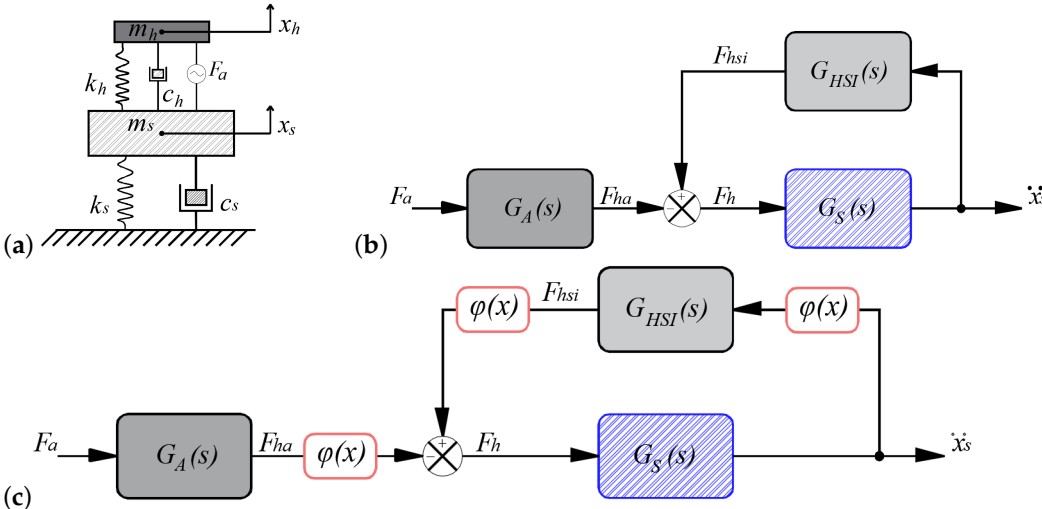

**Figure 1.** (a) Free body diagram for a SDOF structure. (b) Block diagram of the HSI model. (c) Block diagram considering a moving pedestrian.

## 2.2. Passive TMD

The TMD is a passive control device composed of an inertial mass ($m_t$), a viscous damper ($c_t$), and a spring ($k_t$). This device attached to a structure is able to cancel resonant vibrations because its relative movement acts approximately with a delay of 90° with

respect to the motion of the structure, attenuating the structure vibration. The system of equations that governs the behavior of the TMD–structure coupled system is

$$m_t \ddot{x}_t + c_t(\dot{x}_t - \dot{x}_s) + k_t(x_t - x_s) = 0, \tag{8}$$

$$m_s \ddot{x}_s + c_s \dot{x}_s + k_s x_s = F_t - F_{ha} + F_{hsi}, \tag{9}$$

where the subindex $t$ is associated with the modal parameters of the TMD, $F_t$ is the force transmitted by the TMD to the structure, and $F_{hsi} - F_{ha}$ is the external resulting force produced by the human loading. The force $F_t$ is defined from Equation (8) as

$$- m_t \ddot{x}_t = F_t; \longrightarrow c_t(\dot{x}_t - \dot{x}_s) + k_t(x_t - x_s) = F_t. \tag{10}$$

Thus, the TF between the TMD control force and the structure's acceleration is

$$G_T(s) = \frac{F_t(s)}{s^2 X_s(s)} = \frac{-m_t(s c_t + k_t)}{s^2 m_t + s c_t + k_t}. \tag{11}$$

Figure 2 shows the block diagram for the HSI (Figure 1b) model including the feedback loop of the TMD control force. Note that the TMD loop does not depend on the position of the pedestrian action.

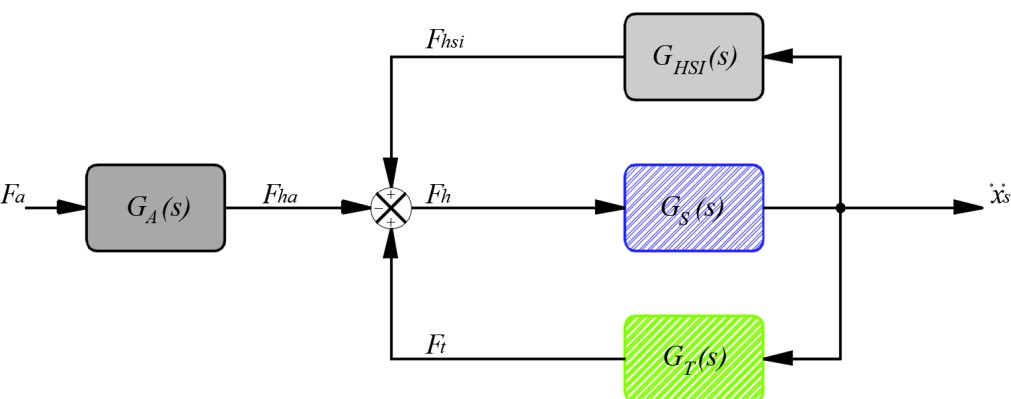

**Figure 2.** Block diagram of the HSI and TMD models.

*2.3. Semiactive TMD*

The STMD makes use of a semiactive damper, which can modify its properties in real-time. For the TMD, perfect tuning occurs with a delay of 90° as mentioned above; however, if the TMD and the structure are not in phase, the nulling vibration effect disappears, adding even more vibration. Hence, an ON–OFF phase control law is assumed for the semiactive damper in such a way that the inertial mass is blocked or released depending on the relative phase between the inertial mass and the structure.

Figure 3 shows the block diagram for the HSI model with a STMD, where the transmitted force from the control device to the structure is obtained from the sum of the elastic force (linear and represented by its TF $G_K(s)$) and damper force (nonlinear and defined by the control law).

Semiactive Control Law

The semiactive control law employed herein consists of an ON–OFF phase control, accounting for the acceleration of the structure and the relative velocity between the structure and the inertial mass to make a decision for the input to the semiactive device. Note that, for lively structures, the relative velocity cannot be approximated by the inertial mass velocity, as was performed in Ref. [12]. Figure 4 shows different states of a SDOF structure controlled by a STMD in order to illustrate the application of the phase control.

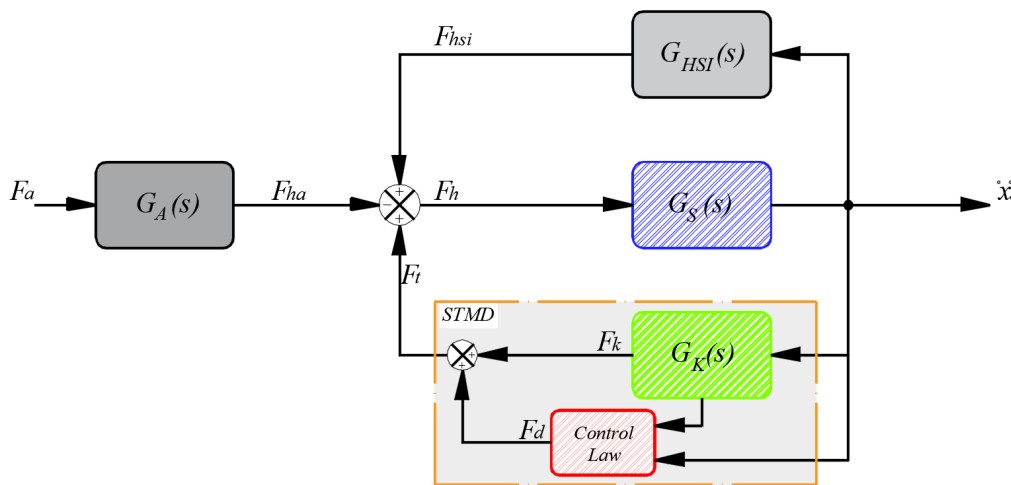

**Figure 3.** Block diagram of the HSI and STMD models.

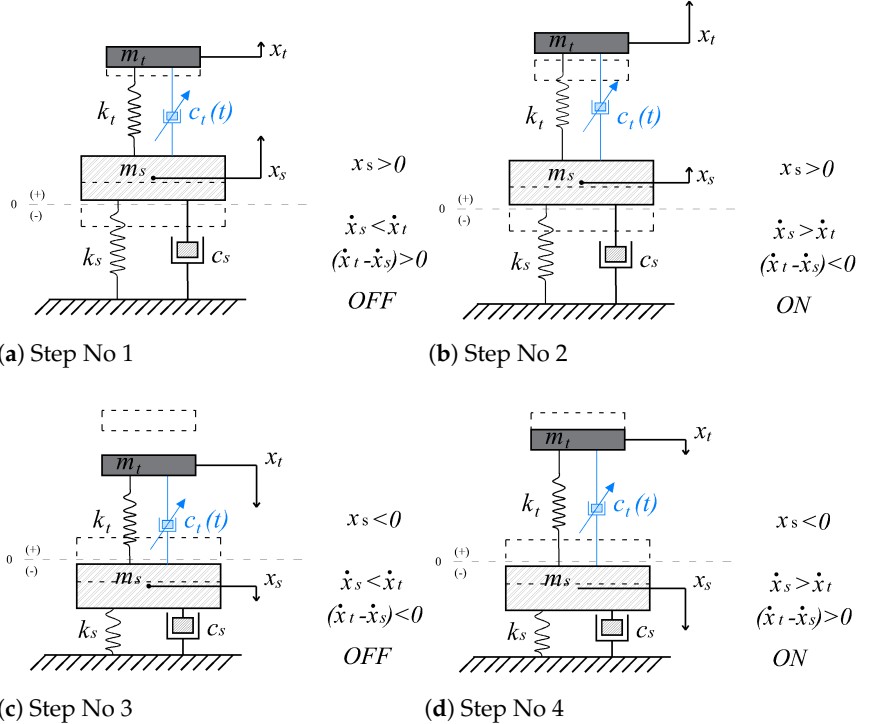

(**a**) Step No 1

(**b**) Step No 2

(**c**) Step No 3

(**d**) Step No 4

**Figure 4.** (**a**,**b**) show the phase control logic for upward motion with respect to an equilibrium state. (**c**,**d**) show the phase control logic for downward motion with respect to an equilibrium state.

The phase control can be expressed by the following inequations:

$$\ddot{x}_s \cdot (\dot{x}_t - \dot{x}_s) \leq 0 \quad \Rightarrow \quad c_t(t) = c_{min} \quad \text{(normal functioning)}$$
$$\ddot{x}_s \cdot (\dot{x}_t - \dot{x}_s) > 0 \quad \Rightarrow \quad c_t(t) = c_{max} \quad \text{(blocking functioning)} \quad (12)$$
$$\text{s.t.:} \ |c_t(t) \cdot (\dot{x}_t - \dot{x}_s)| \leq F_{sat}$$

in which $x_s$ has been substituted by $\ddot{x}_s$ (and changing the inequation sign of Figure 4) and a viscous semiactive damper with force saturation is assumed. The parameters $c_{min}$ and $c_{max}$ are the minimum and maximum damping coefficients, respectively, and $F_{sat}$ is the saturation value for the force.

### 3. Optimum Design Procedure

The proposed methodology consists of the optimum design of a TMD and a STMD by solving a multiobjective constrained optimization problem. The uncertainty associated with the parameters that define the HSI model is taken into account considering $N$ different samples of structures and pedestrian loads. Every structure sample is crossed by one pedestrian whose pacing frequency is tuned to the worst case scenario. Hence, the aim of the optimum design is to obtain an acceleration of the structure below the limit stated by guidelines while minimizing the inertial mass value and the size of the semiactive devices represented by its saturation force.

In Section 3.1, the excitation force are presented. Secondly, the performance indexes and the optimization problem statement are detailed in Section 3.2.

#### 3.1. HSI Model

The TF of the structure including uncertainties (Equation (2)) can be replaced by the following expression:

$$\widehat{G}_S(s) = \frac{s^2 X_s(s)}{F_h(s)} = \frac{s^2}{s^2 \widehat{m}_s + s\widehat{c}_s + \widehat{k}_s},$$

(13)

where "$\widehat{\bullet}$" refers to uncertainty. Although one vibration mode is assumed, the extension to several vibration modes is straightforward by considering their contribution in TF Equation (13).

This must also be applied to the TF associated with the interaction phenomenon (Equation (6)) and that associated with the human load (Equation (7)), yielding

$$\widehat{G}_{HSI}(s) = \frac{F_{hsi}(s)}{s^2 X_s(s)} = \frac{-\widehat{m}_h(s\widehat{c}_h + \widehat{k}_h)}{s^2 \widehat{m}_h + s\widehat{c}_h + \widehat{k}_h},$$

(14)

$$\widehat{G}_A(s) = \frac{F_{ha}(s)}{F_a(s)} = \frac{-s^2 \widehat{m}_h}{s^2 \widehat{m}_h + s\widehat{c}_h + \widehat{k}_h}.$$

(15)

The driving force, $F_a$, which affects the structure and the pedestrian simultaneously, is defined as

$$F_a(t) = Q \left( 1 + \sum_{n=1}^{k} GLF_n \cdot sin(2\pi nft + \psi_n) \right)$$

(16)

where $Q$ is the static load of the pedestrian, $f$ is the pacing frequency, $\psi_n$ is the phase angle of the $n$-th harmonic, $k$ is the number of harmonics considered, and $GLF$ (Generated Load Factor) is the coefficient of the harmonic force associated with the $n$-th harmonic [2]. As mentioned above (Figure 1c), this force is influenced by the pedestrian's position.

#### 3.2. Performance Indexes and the Optimization Problem

The multiobjective optimization problem aims to minimize three objective functions. These are derived from the so-called performance indexes. The multiobjective constrained minimization problem is expressed as follows:

$$\text{find } \boldsymbol{\theta} \text{ such that minimizes } \boldsymbol{\Phi} = (\phi_1, \phi_2, \phi_3)$$

$$\text{s.t. } \begin{cases} \boldsymbol{\theta}_l \leq \boldsymbol{\theta} \leq \boldsymbol{\theta}_u \\ \ddot{x}_{s,max} \leq a_{lim} \end{cases}$$

where $\boldsymbol{\theta}$ are the design parameters; $\phi_{1-3}$ are the three objective functions to be minimized; $\boldsymbol{\theta}_l$ and $\boldsymbol{\theta}_u$ are the lower and upper bounds of the design parameters, respectively; and $\ddot{x}_{s,max}$ is the maximum acceleration of the structure limited to $a_{lim}$. The design variables are the mass, frequency, and damping ratio (TMD) or saturation force (STMD) of the semiactive device for the passive and semiactive version, respectively, so $\boldsymbol{\theta} = (m_t, f_t, \zeta_t \text{ or } F_{sat})$.

The three objective functions, $\phi_{1\text{-}3}$, are defined as

$$\phi_1(z, h, \theta) = w_1 J_{1,P} + w_2 J_{1,RMS}, \quad \text{with} \quad w_1 + w_2 = 1, \tag{17}$$

$$\phi_2(z, h, \theta) = J_2, \tag{18}$$

$$\phi_3(z, h, \theta) = J_3, \tag{19}$$

where $z = [m_s, c_s, k_s]$, $h = [m_h, c_h, k_h]$ are the structure and pedestrian parameters defining the HSI model and $w_1$ and $w_2$ are weighting factors. $J_{1,P}$, $J_{1,RMS}$, $J_2$, and $J_3$ are the performance indexes.

The first index is related to the ratio of maximum peak acceleration of the structure with and without the control strategy implemented (the peak acceleration without control is denoted as $\breve{\ddot{x}}_s$) whereas the second one does the same with the 1-s running root mean square (RMS) acceleration. Thus, the first objective function $\phi_1$ is a balance between the peak acceleration and the persistent acceleration (represented by the RMS acceleration). The third and fourth performance indexes consider the inertial mass and the damping force (TMD) or saturation force (STMD), respectively, which determine the damper size (affecting both the inertial mass and the MR damper needed). Therefore, the four performance indexes are calculated with the following:

- Normalized Peak Acceleration:

$$J_{1,P} = \sum_{i=1}^{N} \left( \frac{max|\ddot{x}_s(t)|}{max|\breve{\ddot{x}}_s(t)|} \right),$$

- Normalized 1s-RMS Acceleration:

$$J_{1,RMS} = \sum_{i=1}^{N} \left( \frac{RMS(\ddot{x}_s(t))}{RMS(\breve{\ddot{x}}_s(t))} \right),$$

- Inertial Mass of the Control Device:

$$J_2 = m_t,$$

- Saturation force:

$$J_3 = F_{sat}.$$

with $N$ being the number of samples. The $F_{sat}$ of the TMD corresponds to the maximum damping force provided by the viscous damper for the whole set of structures and humans considered in the analysis.

## 4. Application of the Proposed Design Methodology

The proposed methodology is applied to the optimum design of a TMD and a STMD to control the excessive vibrations of a real FRP footbridge.

A lightweight FRP footbridge designed using the motion-based design method and constructed at the Laboratory of Structures of the Technical University of Madrid is used as a benchmark structure (see Figure 5). The structure fulfills all the structural limit states except the vibration serviceability one, which should be met through the integration of an inertial damper [2]. The footbridge, which is 10 m-long and 1.5 m-wide, is simply supported at the two ends and consists of three longitudinal Glass FRP (GFRP) stringers connected by transversal GFRP crossbeams placed every 1.20 m. In order to provide a higher bending stiffness and less static sag, Carbon FRP strips are bonded to the top and bottom flanges of the longitudinal stringers. The handrails are comprised of GFRP profiles with a square hollow section connected to the transversal beams. From [2], the first bending mode is achieved at 7.63 Hz, and the second and third mode are torsional modes that are unlikely to be excited by a pedestrian.

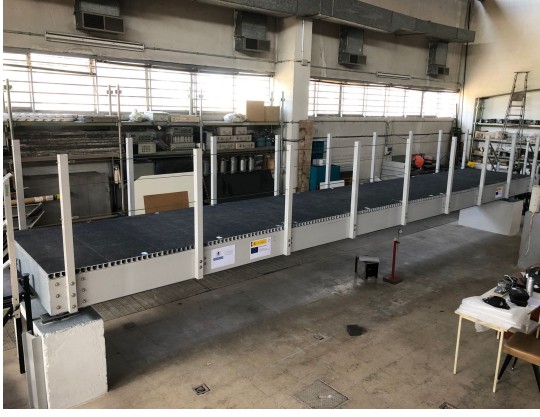
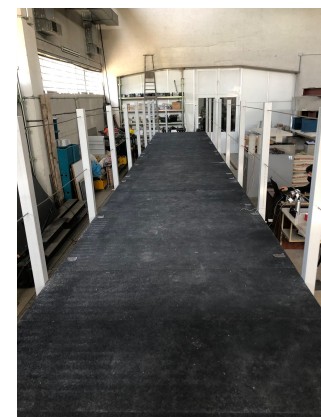

**Figure 5.** FRP footbridge at the Laboratory of Structures of the Technical University of Madrid.

*4.1. HSI Model*

Firstly, $\widehat{G}_s(s)$ is derived using $N = 50$. On the one hand, the generalized modal mass is assumed to follow a normal distribution $\mathcal{N}(405, 0.1^2)$, 405 kg being the mean and 0.1 the standard deviation. On the other hand, the natural frequency and damping ratio are assumed to follow a Weibull distribution, which is recommended to describe the stiffness and strength of pultruded FRP elements [24]. The former is characterized by a scale parameter $\alpha = 8.095$ and a shape parameter $\beta = 7.906$ whereas the latter is characterized by $\alpha = 1.616$ and $\beta = 12.153$. The lower and upper limits for the structure parameters are shown in Table 1. A wide variation range for the structure natural frequency is assumed. This is established on the basis of the following considerations: if the rolling supports are totally blocked, the natural frequency increases by up to 9.13 Hz (that is, under higher rolling friction and or support deterioration, the roller may not be properly activated and the structural natural frequency increases significantly) and if a recycled rubber wearing layer (with a weight up to 60 kg/m$^2$) is placed over the deck, the natural frequency decreases to 5.15 Hz. The frequency response function for each of the $N$ samples is represented in Figure 6, where it can be seen that the detuning effects of the control device could arise when only a nominal structure is considered. From Figure 6, it is concluded that 62% of the uncertain cases present natural frequencies between 6.5 and 8.5 Hz.

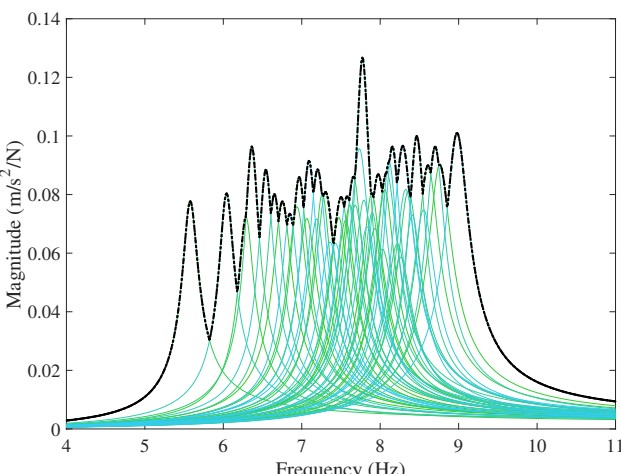

**Figure 6.** FRF behavior for all structures analyzed.

The equivalent properties of the pedestrian, mass and damping ratio, are considered to follow a normal distribution and a uniform distribution, respectively. The normal distribution of the mass is adopted as $\mathcal{N}(65.1, 5^2)$ and the damping ratio is set on the basis of the following limits: the lower limit is equal to 0.1 and the upper limit to 0.4 according to the literature [25]. Hence, based on the previous distributions, the properties of the HSI

model are summarized in Table 1. Note that the frequency of the human body is chosen to be equal to the gait frequency, as suggested by Refs. [2,26,27]. The latter is assumed to be the natural frequency of the structure divided by four in order to affect directly the structure by the fourth harmonic. Therefore, the pedestrian excites the footbridge with up to his/her fourth harmonic. The following parameters, taken from Ref. [2], have been adopted for the definition of the driving force, $F_a$ (Equation (16)): $Q = 686.70$ N; $GLF_1 = 0.1673$, $GLF_2 = 0.1787$, $GLF_3 = 0.1296$, $GLF_4 = 0.0315$, $\psi_1 = 0$, $\psi_2 = -\pi/2$, $\psi_3 = \pi$, and $\psi_4 = \pi/2$. As mentioned in Section 2.1, the MSDA system moves at a velocity depending on the gait frequency according to the expression $v = 1.271f - 1$ (m/s) [28].

**Table 1.** Parameters for the sensitivity analysis.

| | Structure | Value | Units |
|---|---|---|---|
| $m_s$ | Mass | [286.00 to 495.80] | kg |
| $f_s$ | Frequency | [4.60 to 10.40] | Hz |
| $\zeta_s$ | Damping ratio | [0.86 to 1.83] | % |
| $c_s$ | Damping coefficient | [279.40 to 931.10] | kg/s |
| $k_s$ | Stiffness | [3.54 to 17.05] $\times 10^5$ | N/m |
| | **Human** | **Value** | **Units** |
| $m_h$ | Mass | [50.41 to 76.43] | kg |
| $f_h$ | Frequency | $f_s/4$ | Hz |
| $\zeta_h$ | Damping ratio | [10 to 40] | % |
| $c_h$ | Damping coefficient | [164.38 to 643.50] | kg/s |
| $k_h$ | Stiffness | [3.52 to 17.18] $\times 10^3$ | N/m |

*4.2. Optimum Design*

The control devices are designed following the aforementioned methodology to reduce the vibration level of the footbridge. The midspan of the footbridge is considered as a measurement point. The value of $a_{lim}$, which must not be exceeded, is adopted as $a_{lim} = 1$ m/s² for both control devices (medium comfort).

In order to avoid falling into local minimal, a global optimization algorithm is employed to solve the minimization problem. In particular, the multiobjective Genetic Algorithm implemented in the software MATLAB (*gamultiobj* function) is used. An initial population of 50 individuals (parameter vectors with the design variables) are randomly created and iteratively modified according to the natural selection rules of the algorithm, which are based on the selection, crossover, and mutation mechanisms. The selection mechanism selects parents on the current population to create the next generation. Once the parents have been selected, the crossover and mutation mechanism create the new population. The fraction of the population created at the next generation by the crossover function is 0.8. The mutation makes small random changes in the individuals to provide genetic diversity and enable the genetic algorithm to search a broader space. The upper and lower limits of the design variables for the TMD and the STMD are summarized in Table 2. The maximum number of iterations is set from a sensitivity analysis, giving a value of 50 iterations. At each iteration, the objective functions are assessed for the new population. To do this, the objective function runs a Simulink model, which simulates the human–structure–actuator interaction and calculates the maximum acceleration of the footbridge under the human load. When the footbridge acceleration exceeds $a_{lim}$, a penalty is applied to the objective functions.

As a result of the minimization problem, the Pareto front is obtained. Each point forming the curve of the Pareto front represents a possible solution to the problem and the choice of the optimum among all of them may be transformed into a decision-making problem. The three objective functions considered in the problem prevent the classical representation of the Pareto front based on two objectives. Hence, the trade-off point cannot be illustrated and a multidimensional visualization method should be employed. In this

study, the Level Diagrams method proposed by Blasco et al. [29] is used, which allows representing the Pareto front according to proximity to the utopic point on the basis of a given norm. The following steps are carried out.

**Table 2.** Lower and upper bounds for the design variables of the inertial controller.

| | Design Variable | Lower Bound | Upper Bound |
|---|---|---|---|
| $m_t$ | Mass (kg) | 10 | 45 |
| $f_t$ | Frequency (Hz) | 1.00 | 10.00 |
| $\zeta_t$ | Damping ratio (%) | 1 | 50 |
| $F_{sat}$ | Saturation force (N) | 10 | 5000 |

First, the three objective functions are normalized with respect to their maximum and minimum values to between 0 and 1 as follows:

$$\phi_{i,max} = \max\left(\phi_i\right), \qquad \phi_{i,min} = \min\left(\phi_i\right) \quad i = 1, 2, 3$$

$$\bar{\phi}_i = \frac{\phi_i - \phi_{i,min}}{\phi_{i,max} - \phi_{i,min}}. \tag{20}$$

Second, the Euclidean norm (2-norm), defined as $||\bar{\mathbf{\Phi}}||_2 = \sqrt{\sum_i^3 \bar{\phi}_i^2}$, is calculated for the three objective functions selected in this study. The representation of the new Pareto front has a common Y axis representing the Euclidean norm, and each X axis corresponds to the values of the three objective functions. The optimum solution corresponds to the lowest value of the Y axis, i.e., the Euclidean norm. This type of representation of the Pareto front is illustrated in Figure 7 for the TMD and the STMD. Once the method is applied, the optimum parameters of the TMD (mass, frequency, and damping ratio) and the STMD (mass, frequency, and saturation force), displayed in Table 3, are obtained. Note that the required mass ratio for the two devices is higher than the typical mass ratios employed for footbridges constructed with traditional materials. This fact is essentially due to the need to cancel vibrations in a very lightweight structure that may vibrate significantly under nonresonant actions due to the HSI. In case of the TMD, once the optimal parameters $m_t$, $f_t$, and $\zeta_t$ are obtained, the stiffness $k = (2\pi f_t)^2 m_t$ and the damping coefficient $c_t = 2m_t(2\pi f_t)\zeta_t$ are derived. In case of the STMD, once the optimal parameters $m_t$, $f_t$, and $F_{sat}$ are obtained, the stiffness is derived and $c_t$ is calculated from $\zeta = 0.02$. Finally, the control law is implemented using $c_{min} = c_t$ and $c_{max} = 50c_t$.

**Table 3.** Optimum parameters of the TMD and the STMD, and the value of the first objective function.

| | TMD | Value | |
|---|---|---|---|
| $m_t$ | Mass | 33.36 | kg |
| $f_t$ | Frequency | 9.95 | Hz |
| $\zeta_t$ | Damping ratio | 1.03 | % |
| $c_t$ | Damping coefficient | 42.96 | kg/s |
| $k_t$ | Stiffness | $1.30 \times 10^5$ | N/m |
| $\phi_1$ | Objective function | 0.48 | - |
| | **STMD** | **Value** | |
| $m_t$ | Mass | 24.50 | kg |
| $f_t$ | Frequency | 6.15 | Hz |
| $c_{min}$ | Normal functioning | 37.91 | kg/s |
| $c_{max}$ | Blocking functioning | 1895.7 | kg/s |
| $k_t$ | Stiffness | $3.66 \times 10^4$ | N/m |
| $F_{sat}$ | Saturation force | 1010.24 | N |
| $\phi_1$ | Objective function | 0.40 | - |

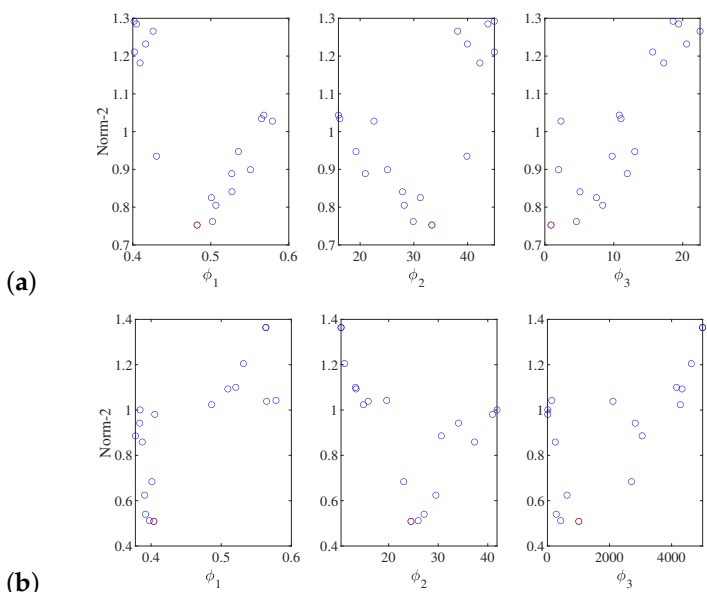

**Figure 7.** Pareto front obtained for the optimum design of the (**a**) TMD and (**b**) STMD. The selected optimum solution is marked in red.

### 4.3. Discussion of Results

From Table 3, $\phi_1$ has similar values in both cases, which indicates similar performance of the controlled structure. However, the inertial mass of the STMD is much lower than the one of the TMD (approximately a 30% reduction). Figure 8a shows a time history example of the dynamic response of the structure when a pedestrian crosses the footbridge. Figure 8b shows the Cumulative Distribution Function (CDF) of not exceeding the 1-s running RMS value and Figure 8c shows the CDF of the instant acceleration for the *N* cases run, giving good evidence of the effect of both control systems under uncertainties. Both control devices fulfill the constraint specified in the optimization algorithm but the STMD outperforms the TMD in terms of the mass (which is significantly smaller) without loss of effectiveness.

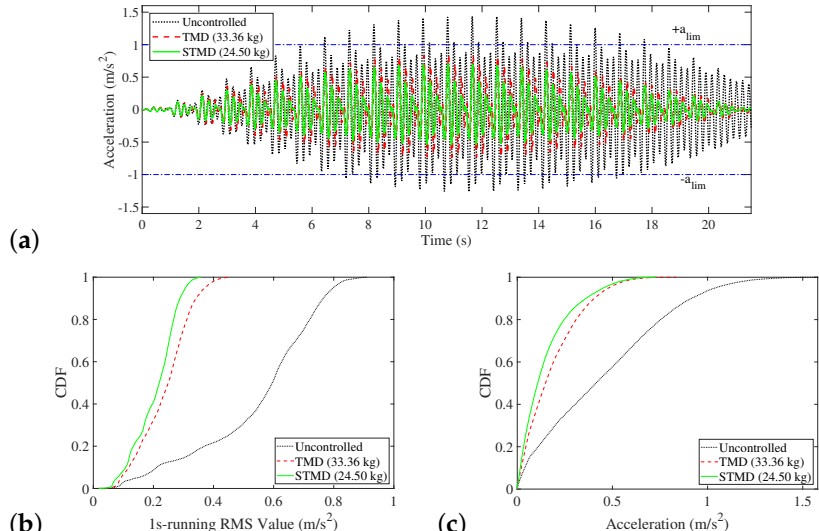

**Figure 8.** Dynamic response of the bare and the controlled structure: (**a**) acceleration at midspan, (**b**) CDF of the 1s-running RMS value of all the samples, and (**c**) CDF of the acceleration of all the samples.

## 5. Application of the Proposed Design Methodology under Realistic Conditions

The proposed controller design methodology is now run under realistic conditions for the STMD. Thus, the following considerations must be taken into account: (i) the viscous semiactive damper model is replaced by a nonlinear MR damper model; (ii) the measured signal used in the control law of the semiactive device will be affected by statistical noise, so that noise-contaminated signals are used; (iii) the low vibration level of the structure could cause unstable behavior of the MR damper since the control law is nonlinear—accordingly, an activation/deactivation rule must be implemented; and (iv) the non-negligible dynamics of filters that are used within the control law implementation are included. At this point, it is worth mentioning that the transient response of the MR damper is not considered since the device responds to voltage changes in milliseconds [30].

### 5.1. MR Modeling

The STMD installed in the structure takes into account the modal parameters obtained in the optimization process, modifying the semiactive viscous device by a MR damper RD-8041-1 of Lord Corporation, which is modeled using a previous identified Bingham model:

$$F - f_o = F_c(V) \cdot sgn(\dot{x}) + c_o(V) \cdot \dot{x}, \tag{21}$$

where the parameters $F$ and $f_o$ are the total force of the MR damper and the preload force, respectively. Note that the preload force may be customized by the manufacturer. In this particular device, it was set to $f_0 = 20$ N. The parameter $F_c$ is the friction force while $c_o$ is the damping coefficient, both dependent on the control voltage. The variable $V$ is the control voltage applied to the MR damper (which is proportional to the current) and, according to the control law described in Equation (12), it can take a minimum or maximum value. In case of the MR damper, these values are 0.5 V and 5 V, respectively. Finally, the velocity $\dot{x}$ corresponds to the relative velocity between the STMD and the structure $(\dot{x}_t - \dot{x}_s)$. The Bingham model used was identified in Barrera et al. [31] and those parameters are used in this section. Figure 9 shows the force–velocity curve of this MR damper for several values of $V$.

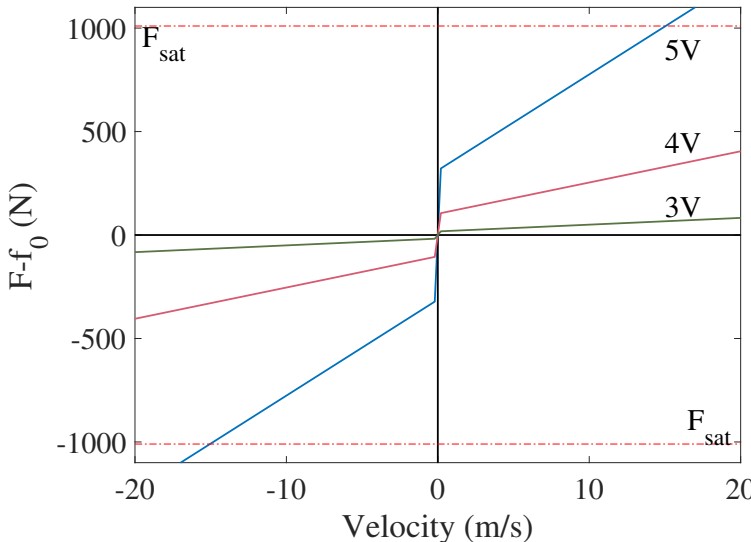

**Figure 9.** Force vs. velocity behavior of the MR damper.

### 5.2. Implementation of the Control Law

In order to account for the presence of noise in the signals, electrical noise has been added to the sensor measurement (structure and inertial mass acceleration, which are generally measured by accelerometers) yielding to noise-contaminated measurements [32]. Additionally, electrical noise is added to the control voltage generated by the control law. These noises are modeled as white noise within a predefined range: a noise peak value

of 0.001 m/s$^2$ is assumed for the structure accelerometer; a noise peak value of 0.01 m/s$^2$ is assumed for STMD mass acceleration; and a noise peak value of 0.02 V is assumed for the control voltage. The peak noise values have been selected according to the available instrumentation to be used in a future implementation.

As commented before, the inertial mass and the structure acceleration are supposed to be measured. Now, the implementation of control law (Equation (12)) considering the MR model (Equation (21)) is explained (and illustrated in Figure 10). Three elements are proposed in the implementation: (i) low-pass filtering of both acceleration signals, (ii) a lossy integrator filter for velocity estimation, and (iii) an activation/deactivation rule.

### 5.2.1. Low-Pass Filter

To remove high-frequency noise and to avoid spillover instabilities, a second-order, low-pass Butterworth filter with a cut-off frequency of 20 Hz is applied to the acceleration used by the control law. Signals with frequencies above the cut-off frequency will be attenuated. The cut-off frequency must be at least twice the frequency of interest; otherwise, the acceleration records will shift in its phase, affecting the control law performance significantly. The low-pass Butterworth filter with a sampling frequency of 1000 Hz (the one used for numerical simulation and the expected one for the experimental implementation) is used and its Z-transform is as follows:

$$H_{low}(z) = \frac{0.0036z^2 + 0.0072z + 0.0036}{z^2 - 1.823 + 0.837}. \tag{22}$$

### 5.2.2. Integrator Filter

The relative velocity between the structure and the inertial mass is obtained by the integration of the acceleration signals. In Section 4, an ideal integrator $H_{int}(s) = 1/s$ was used. However, this type of integrator cannot be used experimentally since it is extremely sensitive for low-frequency components in real implementation [33]. A lossy integrator can maintain the magnitude and phase of an ideal integrator from a cut-off frequency, avoiding the high sensitivity at low frequencies, and removing signal offsets. Thus, the lossy integrator could be expressed through the following equation:

$$H_{int}(s) = \frac{s}{s^2 + 2s\zeta_i\omega_i + \omega_i^2}, \tag{23}$$

where $\zeta_i$ is the damping ratio of the integrator, which is chosen equal to 1, and $\omega_i$ is the cut-off frequency, which should be sufficiently smaller than the frequencies of interest in order to maintain the magnitude and phase. A cut-off frequency of $2\pi f_s/10$ has been adopted here. As in Equation (22), the Z-transform for a sampling frequency of 1000 Hz is derived:

$$H_{int}(z) = \frac{0.00099z - 0.00099}{z^2 - 1.99z + 0.99}. \tag{24}$$

### 5.2.3. Deactivation Rule

To prevent the MR damper from operating at low acceleration values, a deactivation rule has been adopted. This situation may occur under ambient loads. The application of ON/OFF control under low input signals may lead to instabilities (chattering problems) that are avoided by disconnecting the device. Additionally, the electrical consumption is reduced and the device lifespan is extended. For this case, the OFF state is applied for 1-s-RMS instant acceleration values of the structure lower than the threshold of 0.01 m/s$^2$, so the input voltage to the MR damper is the minimum possible indicated by the manufacturer.

Figure 10 illustrates the block diagram of a structure with a HSI model and a semi-active control strategy. The noise in sensors, the low-pass filter, integrator filters, and the activate/deactivate rule have been included to represent a realistic scenario.

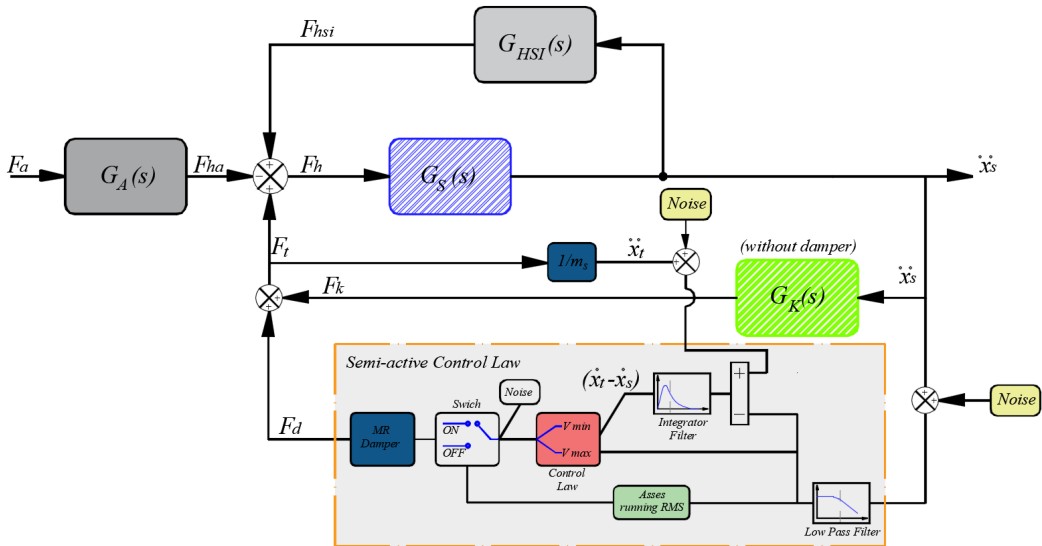

**Figure 10.** Block diagram of the HSI and STMD including all elements of the implementation and noisy signals.

### 5.3. Optimum Design under Realistic Conditions

The application of the proposed design methodology is carried out again considering the MR model, noise-corrupted signals, and all the dynamics of the elements needed to experimentally implement the control law. Table 4 shows the optimum values obtained for the STMD MR (which includes all components mentioned before). The optimal values obtained are similar to those derived in Section 4 (Table 3, but the objective function $\phi_1$ shows a reduction in the global performance of the STMD). In Figure 11, the three optimum controllers (TMD, STMD, and STMD MR) are compared. Figure 11a shows the time history of acceleration for a particular case. Figure 11b,c show the CDF for the 1-s-running RMS and instant acceleration for the N cases run, respectively. It can be observed that the STMD MR maintains the maximum values of acceleration under $1 \, \text{m/s}^2$, fulfilling strictly the constraint of the optimization problem. Three observations can be obtained at this point: (i) for the peak acceleration, the TMD and the STMD MRs show similar performances with a significant reduction in the inertial mass of the STMD (contributing thus to the lightness of the footbridge); (ii) for the TMD, it should be noted that ideal conditions have been assumed (dry friction, misaligned of the springs, nonlinear behavior of the viscous damper, and other experimental issues might reduce its performance) so a lower performance of the TMD is expected; (iii) the STMD MR slightly reduces its performance compared with the viscous STMD, as expected.

**Table 4.** Optimum parameters of the STMD MR.

| STMD MR | | Value | |
|---|---|---|---|
| $m_t$ | Mass | 24.38 | kg |
| $f_t$ | Frequency | 6.00 | Hz |
| $k_t$ | Stiffness | $3.47 \times 10^4$ | N/m |
| $F_{sat}$ | Saturation force | 88.27 | N |
| $\phi_1$ | Objective function | 0.48 | - |

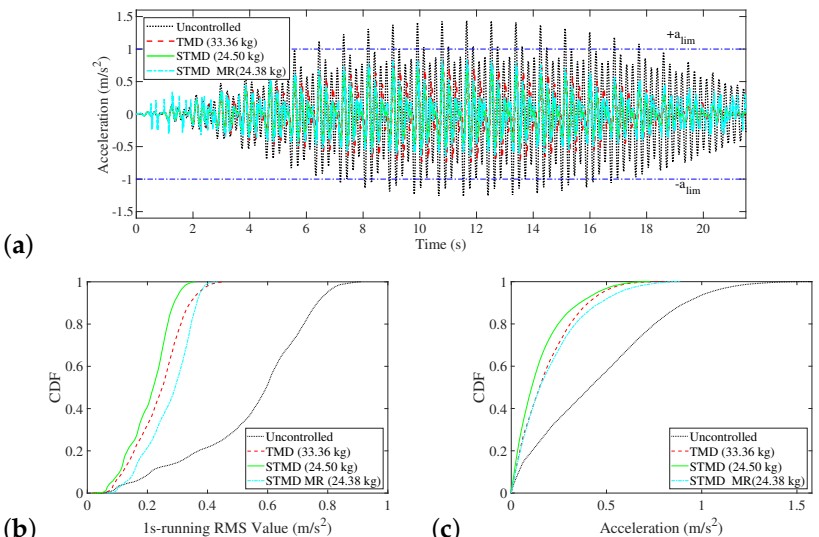

**Figure 11.** Dynamic response of the footbridge with each control device: (**a**) Acceleration at midspan; (**b**) CDF of the 1-s-running RMS value of all the samples; (**c**) CDF of the acceleration of all the samples.

Finally, the performance of the two STMDs is evaluated in terms of the objective function $\phi_1$ under a real walking ground reaction force—that is, in Figure 10, $F_{ha}$ is now the vertical reaction force measured in an instrumented treadmill so that the input is not a pure harmonic action (see Figure 12). For this purpose, the footbridge cases with natural frequencies equal to 5.48, 7.63, and 10.39 Hz are considered. The natural frequency of 7.63 Hz corresponds to the natural frequency of the constructed footbridge (see Section 4). The results are shown in Figure 13 and it is clearly observable the improvement obtained with the implementation of the STMDs, which achieves a substantial reduction of the acceleration. For the two cases analyzed under real walking excitation, it is shown how the performance of the STMD MR is degraded with respect to the viscous one, although this fact may be improved by implementing a continuous control law in future works.

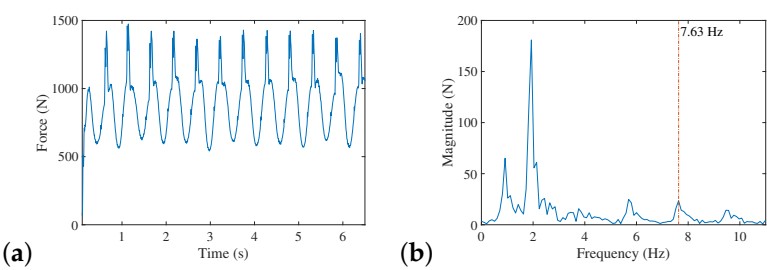

**Figure 12.** Vertical reaction force measured in an instrumented treadmill: (**a**) Time history; (**b**) Fast Fourier Transform.

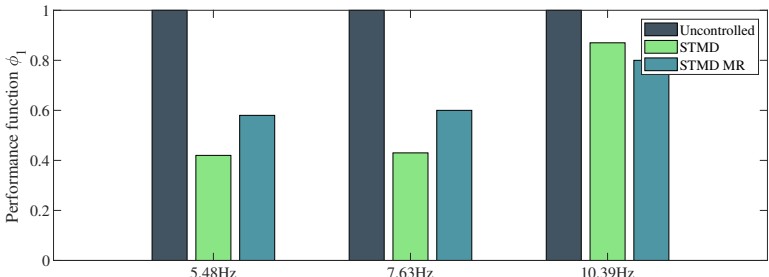

**Figure 13.** Objective function $\phi_1$ of the optimum systems under different footbridge frequencies.

## 6. Conclusions

A methodology for the optimum design of a STMD using a semiactive damper controlled by a phase-control law has been proposed in this study. The methodology framework included, for the first time, the human–structure–actuator interaction. The design process was based on the solution of a multiobjective constrained minimization problem in which the parameters that define the control device are the design variables of the problem. The constraint has been adopted from the maximum vertical acceleration allowed by guidelines. This methodology has been applied to the optimum design of a STMD for reducing the dynamic response of a lightweight FRP footbridge, obtaining a vibration reduction of more than 50% with respect to the uncontrolled case. To take into account the uncertainties of the coupled human–structure system, 50 samples of the structure and the equivalent human system were generated. The same case was used to compare the STMD with the passive TMD. Thus, a similar reduction capacity was achieved with both devices but a 26% higher mass was required by the passive TMD.

Finally, the optimization process for a realistic simulation of the STMD has been undertaken, using an MR damper (modeled by a Bingham model) as a semiactive device and including the required elements that must be taken into account for the feedback process and the implementation of the control law. The results reported a performance reduction compared with the TMD and the ideal STMD. Although an ON/OFF phase control has been adopted here, the proposed design methodology can be applied for other control strategies, as long as the phase adjustment procedure corresponds to the one explained in Section 2.3.

Future works will be focused on the experimental development of the methodology proposed herein on the FRP footbridge constructed at the Laboratory of Structures of the Technical University of Madrid. Besides, strategies for reducing the influence of the MR nonlinear model will be explored such as dynamics inversions and/or continuous control laws.

**Author Contributions:** Conceptualization, C.A.B.-V. and I.M.D.; methodology, I.M.D.; software, C.A.B.-V. and J.N.-P.; validation, I.M.D. and J.H.G.-P.; formal analysis, J.H.G.-P.; investigation, C.A.B.-V. and J.N.-P.; data curation, I.M.D. and J.H.G.P.; writing—original draft preparation, C.A.B.-V. and J.N.P.; writing—review and editing, I.M.D. and J.H.G.-P.; funding acquisition, I.M.D. All authors have read and agreed to the published version of the manuscript.

**Funding:** The authors acknowledge the Grant RTI2018-099639-B-I00, "Structural efficiency enhancement for bridges subjected to dynamic loading: integrated smart dampers", funded by MCIN/AEI/ 10.13039/501100011033 and by "ERDF A way of making Europe". Christian A. Barrera-Vargas want to thank the FPI PRE2019-087494 predoctoral grant.

**Institutional Review Board Statement:** Not applicable.

**Informed Consent Statement:** Not applicable.

**Data Availability Statement:** Not applicable.

**Acknowledgments:** The authors would like to thank Christian A. Gallegos and José M. Soria for their support and sharing their knowledge during the development of this research.

**Conflicts of Interest:** The authors declare no conflict of interest.

## Abbreviations

The following abbreviations are used in this manuscript:

| | |
|---|---|
| FRP | Fiber-Reinforced Polymers |
| TMD | Tuned Mass Damper |
| STMD | Semiactive Tuned Mass Damper |
| HSI | Human–Structure Interaction |
| MSDA | Mass-Spring-Damper-Actuator |
| MTMD | Multiple Tuned Mass Damper |

MR      Magnetorheological
DLF     Dynamic Load Factors
GLF     Generated Load Factors
SDOF    Single Degree of Freedom
CDF     Cumulative Distribution Function

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
