# Peer review of "Design of a Semiactive TMD for Lightweight Pedestrian Structures Considering Human–Structure–Actuator Interaction"

_actuators, doi:10.3390/act11040101_

Round 1

Reviewer 1 Report

This is an interesting paper focused on a relevant topic of STMD design to mitigate pedestrian-induced vibrations in footbridges. I have few comments and recommendations.

1) Introduction - other control algorithms for STMDs, not only the on-off, should be mentioned. In this paper, the on-off control strategy [8] is assumed but the presented design approach can be used regardless of the control strategy, right?

2) What function in MATLAB is used for the optimization, is it gamultiobj? During optimization, is the maximum structure acceleration determined by simulation of the Simulink model? The optimization procedure should be better described.

3) From where the Authors take the values of Q, GLF (page 9, line 267)?

4) The Bingham model is used in the numerical study but not in the optimization. Why is it not used already during optimization? 

5) What is the value of f0 in the Bingham model (fig. 9)? What will be the best value of f0 (and also Vmin) from the perspective of the STMD performance? 

6) It should be mentioned that the "realistic simulation", as called by the Authors, should also take into account the dynamic properties of the on-off controlled MR damper, at least its response time which influences the performance of the MR damper-based STMD (note: I suggest minor correction - it is not required to include this analysis in this paper, but it's needed to be mentioned).

Author Response

Dear Reviewer. 

I would like to thank you for your comments and suggestion. We're attaching the PDF document with the responses point by point. 

Thank you very much for your attention. We hope that the updated version of the manuscript solved all hesitates. 

Best regards,
Iván M. Díaz

Reviewer 2 Report

The paper presents an interesting methodology for designing STMDs intended to mitigate the human induced vibration of slender structures including the Human-Structure interaction (HSI) phenomenon. It is based on a multi-objective optimization which includes the HSI model together with the semi-active control law. Although based on a real structure, the provided case study and examples do not show experimental results but only numerical ones.

The topic is of great interest, since this approach can be considered as an evolution of the TMD in order to increase its robustness, but the paper cannot be published in its current form due to several issues. Please refer to the attached document for more details.

Author Response

Dear Reviewer. 

I would like to thank you for your comments and suggestion. We're attaching the PDF document with the responses point by point. 

Thank you very much for your attention. We hope that the updated version of the manuscript solved all hesitates. 

Best regards

Reviewer 3 Report

The paper is well written and very clear in all its parts. I am happy to support its publications after some of my questions are answered:

  1. It is not totally clear the declaration of novelty especially that the novelty is related to the model used. For example what is the novelty with respect to your reference [9]?
  2. When you introduce the SDOF model it would be good to justify it (at the moment the justification is later in the paper)
  3. Four performance indices are used in the optimization problem. Please justify the choice of such particular indices.
  4. In figure 8a the STMD seems marginally improving over the use of a TMD. It is not clear if the added complexity of a STMD is totally justifiable. I think this should be clarified and the performance of a TMD added to table 4. 

Author Response

(The authors gave the same response as above.)

Round 2

Reviewer 2 Report

I would like to sincerely acknowledge the authors for answering all my comments and modify the article accordingly. The quality of the work has substantially improved and it can be published in its present form.I

Author Response

Thank you very much for your positive comment.